# Different Flooding Conditions Affected Microbial Diversity in Riparian Zone of Huihe Wetland

**DOI:** 10.3390/microorganisms13010154

**Published:** 2025-01-13

**Authors:** Bademu Qiqige, Jingjing Liu, Ming Li, Xiaosheng Hu, Weiwei Guo, Ping Wang, Yi Ding, Qiuying Zhi, Yuxuan Wu, Xiao Guan, Junsheng Li

**Affiliations:** 1China Geological Survey Comprehensive Survey Command Center for Natural Resources, Beijing 100055, China; bdmqqg@126.com (B.Q.); lm18910077797@163.com (M.L.); et_et@126.com (X.H.); guoweiwei0916@163.com (W.G.); wangping@bjfu.edu.cn (P.W.); dingyi01@mail.cgs.gov.cn (Y.D.); 2Chinese Research Academy of Environmental Sciences, Beijing 100012, China; liujj961208@163.com (J.L.); 220220932760@lzu.edu.cn (Q.Z.); wuyuxuan221@mails.ucas.ac.cn (Y.W.); 3College of Ecology, Lanzhou University, Lanzhou 730020, China

**Keywords:** wetland, plant, soil physicochemical properties, bacteria, fungi, archaea

## Abstract

The soil microbiome plays an important role in wetland ecosystem services and functions. However, the impact of soil hydrological conditions on wetland microorganisms is not well understood. This study investigated the effects of wetted state (WS); wetting–drying state (WDS); and dried state (DS) on the diversity of soil bacteria, fungi, and archaea. The Shannon index of bacterial diversity was not significantly different in various flooding conditions (*p* > 0.05), however, fungal diversity and archaeal communities were significantly different in different flooding conditions (*p* < 0.05). Significant differences were found in the beta diversity of bacterial, fungal, and archaeal communities (*p* < 0.05). Additionally, the composition of bacteria, fungi, and archaea varied. Bacteria were predominantly composed of *Proteobacteria* and *Actinobacteria*, fungi mainly consisted of *Ascomycota* and *Mucoromycota*, and archaea were primarily represented by *Crenarchaeota* and *Euryarchaeota*. Bacteria exhibited correlations with vegetation coverage, fungi with plant diversity, and archaea with aboveground vegetation biomass. The pH influenced bacterial and archaeal communities, while soil bulk density, moisture, soil carbon, soil nitrogen, and plant community diversity impacted fungal communities. This study provides a scientific basis for understanding the effects of different hydrological conditions on microbial communities in the Huihe Nature Reserve; highlighting their relationship with vegetation and soil properties, and offers insights for the ecological protection of the Huihe wetland.

## 1. Introduction

The wetland ecosystem is the core of the earth’s ecosystem, which is the transition zone between water and land [1]. Wetlands have both abundant aquatic and terrestrial plant and animal resources, forming a natural gene pool and having a unique biological environment compared with any other single ecosystem [2]. The complex environment and climate conditions have resulted in multitudinous creatures, which have irreplaceable ecological value for maintaining biodiversity and ecological balance [3]. The riparian zone (riparian buffer) is the main part of wetland ecosystems and is an amphibious ecotone formed by the transition from aquatic ecosystems to terrestrial ecosystems [4,5]. The riparian zone has a specific geographic location and varied environmental conditions; as the riparian zone is the most active area, it can transmit and convert material, energy, and information between aquatic and terrestrial ecological environments, with both water and land characteristics [6]. For example, nitrogen (N) cycling is different in riparian zones than in other ecosystems, in which both energy flow and nutrient cycling are more active than in adjacent aquatic and terrestrial areas [7]. The river water level fluctuates periodically in different seasons, and the riparian zone can alternate between dry and wet conditions; that is, the alternating process of drying and wetting impacts the vegetation and soil microbial communities [8]. Vegetation and soil play key ecological functions in riparian zones, increasing biodiversity, reducing soil erosion, storing carbon, and facilitating climate regulation [9]. Riparian zones possess abundant nutrients through high production and nutrient cycling, and they are anoxic microsites that have a high rate of denitrification [10]. Soil microbes transform available nitrate into di-nitrogen gas at a high rate by denitrification at anoxic microsites (mostly riparian) [11]. The soil environment is complicated and dynamic in the riparian zone; as an integral part of the riparian zone, soil microbes promote the decomposition and transformation of soil organic carbon [12] and maintain ecosystem function simultaneously, while plants and the soil environment drive the community composition and diversity of soil microbes [13].

Specific soil microbial communities and functional groups exist in the environments of riparian zones, and they are related to specific soil characteristics in such zones [14]. Soil microbial communities are relatively sensitive to soil physicochemical conditions, including soil pH, soil moisture, and available N [15], whereas soil moisture strongly affects fungal alpha biodiversity. Seasonal changes in the water level of aquatic ecosystems constantly alter the soil environment in riparian zones [16,17]. Seasonal variation, caused alterations in the water level, further disturb the soil microbial community (i.e., flooding or drought) [18]. The soil water content affects the physiological and biochemical properties of bacteria and archaea and restricts the decomposition of certain types of soil organic matter [19]. Flooding or drought can alter anoxic and oxic conditions, benefit nitrification and denitrification, and further affect the activity of microbes [20]. In recent decades, most of the studies of riparian zones have focused primarily on the relationships between vegetation communities and soil microbes affecting carbon sequestration and aggregate stability [21]. In the riparian zone, the ability of plant species to promote soil N and P mineralization was greater than that of plant biomass, and vegetation cover, regardless of the plant species, improved available nutrient supplies and retention and shaped the soil microbial community [22].

Interactions between the vegetation community structure and soil microbes affect soil aggregates, water-holding ability, and N mineralization [23]. Riparian zones are specific environmental conditions that can provide adequate nutrients for vegetation and soil creatures, which further leads to community succession and strengthens the functionality of the ecosystems in riparian zones [24]. To date, studies have focused primarily on the release of N_2_O during nitrification or denitrification processes in riparian zones [25], and the spatial changes in soil microbes in alternating wet and dry environments in the riparian areas are still unclear.

The Huihe wetland is the largest band-shaped herbaceous wetland in Inner Mongolia; it is located in the eastern part of the grassland and constitutes the grassland wetland ecosystem in Northeast Asia, with three main parts (Dalai Lake in Hulunbuir City, Davursk in Russia and the Daur Wetland in Mongolia). The Huihe wetland is a vital ecological barrier in Northeast Asia and even the whole world, as well as an important line of ecological defense in northern China. The Huihe wetland plays an important role in maintaining the ecological balance of North China as a whole. The Huihe wetland is in the semiarid climate area in the southern part of the Hulunbuir Grassland, and the ecosystem of the Huihe wetland is relatively fragile. In recent years, many challenges have been faced, such as vegetation degradation and ecological service function decline. We wanted to study the changes in vegetation, soil, and microbes in the Huihe wetland and further explore the coupling relationships among plants, soil, and microbes in the wetland. Here, we hypothesize that (i) in different flooded areas, the diversity and community construction of bacteria, fungi and archaea are inconsistent, and the bacteria are relatively stable, which may not easily change with the environment; thus, their diversity may not differ, and the diversity of fungi and archaea varies with the environment under different flooded conditions. (ii) Changes in environmental factors, such as various water conditions and soil nutrients, significantly affect the characteristics of bacterial, fungal, and archaeal communities, and their driving factors differ.

In this study, we investigated vegetation communities and collected soil samples from the wetlands in the Huihe region, extracted all the DNA from the samples, amplified them via PCR, quantified them, and conducted a primary study on the different microbes (soil bacteria, fungi, and archaea) via a metagenomic sequencing platform. Our study aimed to analyze the complex relationships among vegetation properties, soil microbial communities, and soil physicochemical properties, and the results reveal a microbial ecological perspective for understanding the response of soil microbes to environmental physicochemical properties in the core area of the riparian zone (WS, WDS, and DS) of the Hui River National Nature Reserve.

## 2. Materials and Methods

### 2.1. Study Site Description

This study was conducted at the Huihe River National Nature Reserve, Ewenki Autonomous Banner, Inner Mongolia Autonomous Region (Figure 1), China (119°16′22″ E~119°41′39″ E, 48°10′50″ N~48°56′11″ N; 600~1000 m altitude). The wetland covers an area of 1.167 × 10^3^ km^2^. The climate is a mid-temperate continental monsoon climate with an annual rainfall of 300–350 mm, more than 70% of which is concentrated from July–August. The annual mean temperature is −2.4 to 2.2 °C. According to the World Reference Base for Soil Resources (WRB) classification system [26], the soil type in the study area belongs to Histosols. The vegetation community is dominated by *Cleistogenes squarrosa*, *Koeleria cristata*, *Stipagrandis*, *Leymus chinensis*, and *Artemisia frigida* [27].

### 2.2. Experimental Design

From July to September 2021, in the Huihe National Nature Reserve, Ewenki Autonomous Banner, 12 plots were set along the upper and lower reaches of the Huihe River (Figure 1), and each plot was subjected to three types of folding treatments: long-term flooding (wetted state, WS), a wetting–drying state (WDS), and a dried state (DS). Every condition included 5 random replications, and we collected plant and soil samples from each experimental plot.

### 2.3. Plant and Soil Sampling

In 2021, there were 12 plots set in the Huihe River, and each plot had 3 kinds of conditions (WS, WDS, and DS). The plant community characteristics were surveyed in five 1 m × 1 m random quadrats per treatment along every transect. A total of 180 plant quadrats were investigated, and each quadrat investigated the total number of plant species, the coverage of plants, and aboveground biomass. After researching the plant species, we collected surface soil samples from depths of 10 cm for physical, chemical, and metagenome analyses of the soil in the same quadrat, and five random soil samples were collected in every quadrat using a soil auger. All the soil samples from each quadrat were mixed completely into one sample, which was prepared for the determination of related indicators. The mixing of multiple samples can reduce the error range, and is used in many studies for measuring the soil’s physical and chemical properties and soil microbe content. A total of 180 soil samples were collected.

### 2.4. Plant Properties and Soil Physicochemical Properties

The plant species diversity was calculated by using the following formula [28]: $IV=(RH_{i}+RC_{i})/2$, where *RC* represents the relative cover and *RH* represents the relative height of the community of the aboveground biomass (AGB). First, the soil samples were sieved by a sieving mash (2 mm) to remove redundant impurities such as stones and plant roots. The soil samples were separated into three parts: one subsample for the determination of different analyzed indicators, one subsample for the determination of soil properties, which was dried at room temperature (about 25 °C), and the other subsample for metagenomic sequencing at −80 °C.

We followed methods to determine the total carbon (TC), total phosphorus (TP), soil total nitrogen (TN), soil organic carbon (SOC), soil water content (SWC), soil pH, and electrical conductivity (EC) [29]. The surface soil at a 0–20 cm depth was measured for the soil bulk density (SBD) by the cutting ring (100 cm^3^) method [30], and the coverage and aboveground biomass were observed and measured in a quadrat.

### 2.5. Metagenomic Sequencing of the Soil Samples

#### 2.5.1. DNA Extraction and PCR Amplification

The CTAB method was employed to extract DNA from the soil samples. DNA quality, including degradation level, possible contaminants, and concentration, were assessed using an Agilent 5400 system. For sequencing, the NEBNext^®^ UltraTM DNA Library Prep Kit for Illumina (NEB, Ipswich, MA, USA, Catalog#: E7370L) was utilized. In summary, DNA was fragmented via sonication (350 bp), followed by end-repair, A-tailing, adapter ligation for Illumina sequencing, and PCR amplification. The resulting PCR products were further purified using the AMPure XP system (Brea, Beverly, CA, USA). Library quality was again checked on the Agilent 5400 system and quantified via QPCR. The libraries were then sequenced on Illumina platforms using the PE150 strategy.

#### 2.5.2. Data Quality Control and De-Host Sequences

The raw data of the microbes (bacteria, archaea and fungi) in the soil samples were acquired via metagenomic sequencing (Illumina NovaSeq, a high-throughput sequencing platform). To confirm the high quality of the data, the raw sequencing data were prepossessed with Kneaddata software (0.12.0). (1) Sequencing adapters and low-quality bases were removed, and sequences with final lengths less than 50 bp (parameter MINLEN: 50) were removed via Trimmomatic. (2) Considering the possibility of host contamination in the samples, Clean Data needed to be the host database, which used Bowtie2 software (2.3.5.1. http://bowtie-bio.sourceforge.net/bowtie2/manual.shtml accessed on 9 December 2024) to filter for subsequent analysis. (3) Finally, FastQC was used to detect the rationality and effects of quality control [31]. Species samples were identified using Kraken2 (v2.1.3), in conjunction with a custom microbial database comprising sequences from bacteria, fungi, and archaea. This dataset was compiled using information from the NT nucleic acid database and the RefSeq complete genome database, which was provided by NCBI. Following species identification, Bracken was employed to determine the accurate relative abundance of each species within the samples.

### 2.6. Statistical Analysis Method

The Shannon, Simpson, and species richness indices were used to assess the plant community α-diversity, and the diversity of the soil microbial community was assessed using the Chao, Shannon, and Sobs indices. One-way ANOVA was used to analyze the Shannon diversity, soil physicochemical properties, the relative abundances of various microbes, α-diversity indices, and the microbe taxes from the different flooding treatments. The β-diversity of the soil microbes was analyzed via nonmetric multidimensional scaling (NMDS) analysis [32]. We identified classification levels for bacteria at the phylum level, fungi at the family level, and archaea at the family level. A multiple regression linear model and variance decomposition analysis were applied to calculate the proportion of explanatory variability, which analyzed the biological contributions of the core microbiota to the soil physicochemical properties and the total explanatory ability of the corresponding physical and chemical properties of the core microbes. A structural equation model (SEM) was applied to test the relationships between the soil microbial diversity (bacterial, fungal, archaeal) and environmental factors [33]. R 4.0.2 for Windows was used to analyze all variables and draw the figures, and the “vegan (2.6.8)”, “psych (2.4.6.26)”, and “piecewiseSEM (3.5.0)” packages were used to apply the SEM.

## 3. Results

### 3.1. Soil Microbial Biodiversity and Community Construction

The bacterial Shannon index was not significantly different (*p* < 0.05) among the long-term flooding (wetted state, WS), wetting–drying state (WDS), and dried state (DS) samples, and the Shannon index ranged from 4.8–5.1 (Figure 2A). However, the fungal diversity of DS was significantly greater than that of the WS and WDS soils, which were DS > WDS > WS (*p* < 0.05) (Figure 2B). While archaea presented the opposite trend to fungi with increasing moisture, the archaeal Shannon index of DS was significantly greater than that of WS (Figure 2C). NMDS analysis revealed that the soil microbial samples from WS, WDS, and DS soils formed distinct clusters within the ordination space (Figure 3A,C,E), confirming significant differences at the taxonomic level via the ANOSIM test. These differences among the WS, WDS, and DS soils were significant in terms of the bacterial, fungal, and archaeal communities, which indicates that the soil’s fungal and archaeal communities were easily disturbed by the different flooding water levels. Furthermore, we assessed the variation in beta diversity among the bacterial, fungal, and archaeal community groups on the basis of the Bray–Curtis distance (Figure 3B,D,F) and detected significant differences in beta diversity, which indicated greater dispersion. At the phylum level, Actinobacteria, Proteobacteria, *Acidobacteria*, *Cyanobacteria*, and *Firmicutes* dominated the bacterial community under different flooding conditions. *Ascomycota*, *Basidiomycota*, *Chytridiomycota*, and *Mucoromycota* were the dominant organisms in the fungal community (phylum level). *Candidatus Bathyarchaeota*, *Candidatus Diapherotrites*, and *Candidatus Thermoplasmatota* dominated the archaeal community under different flooding conditions (Figure 4A–C).

### 3.2. Key Factors Driving Soil Microbial Communities

To identify the (primary) environmental factors driving soil microbes, we correlated the distance correction differences in bacterial, fungal, and archaeal community compositions with the differences in environmental factors. The Mantel test revealed that plant cover was the driving factor of the bacterial, fungal, and archaeal communities (Figure 5). Bacteria were closely correlated with pH, and fungi were linked with SBD, SWC, TC, TN, and the Shannon index (Table 1). Archaea was strongly correlated with AGB. Furthermore, multiple regression modeling was used to estimate the biological associations of the dominant microbes (bacteria, fungi, and archaea) at the phylum level with variations in soil physicochemical properties. We found that different microbial phyla contributed to the soil physicochemical properties in WS, WDS, and DS soils. For example, *Crenarchaeota* abundance was a vital variable for predicting plant properties in WS, including plant cover and AGB; however, this level did not contribute to the plant cover or AGB in WDS or DS soils. This result indicates the importance of *Crenarchaeota* in the plant properties of WS rather than those of WDS and DS. Other vital variables for predicting soil physicochemical properties in WS were the *Thaumarchaeota* abundance for pH, EC, and SBD; the *Crenarchaeota* abundance for plant cover and AGB; the *Proteobacteria* abundance for SWC, TC, SOC, TP, and TN; the *Ignavibacteriae* abundance for pH and EC; the *Firmicutes* abundance for EC and SBD; the *Chloroflexi* abundance for pH; the *Actinobacteria* abundance for EC; and the *Acidobacteria* abundance for AGB. Other important variables for predicting soil physicochemical properties in WDS soils were *Thaumarchaeota* and *Chloroflexi* abundance in terms of pH; *Proteobacteria*, *Firmicutes*, *Euryarchaeota*, *Firmicutes*, *Cyanobacteria*, and *Actinobacteria* abundances in the SBD; and *Planctomycetes* abundance in the pH and SBD. The abundances of *Tenericutes*, *Ignavibacteriae*, *Cyanobacteria*, and *Actinobacteria* were associated with pH; *Spirochaetes* and *Proteobacteria* abundances were associated with EC; and *Euryarchaeota* abundances were associated with pH and EC (Figure 6).

**Figure 3 microorganisms-13-00154-f003:**
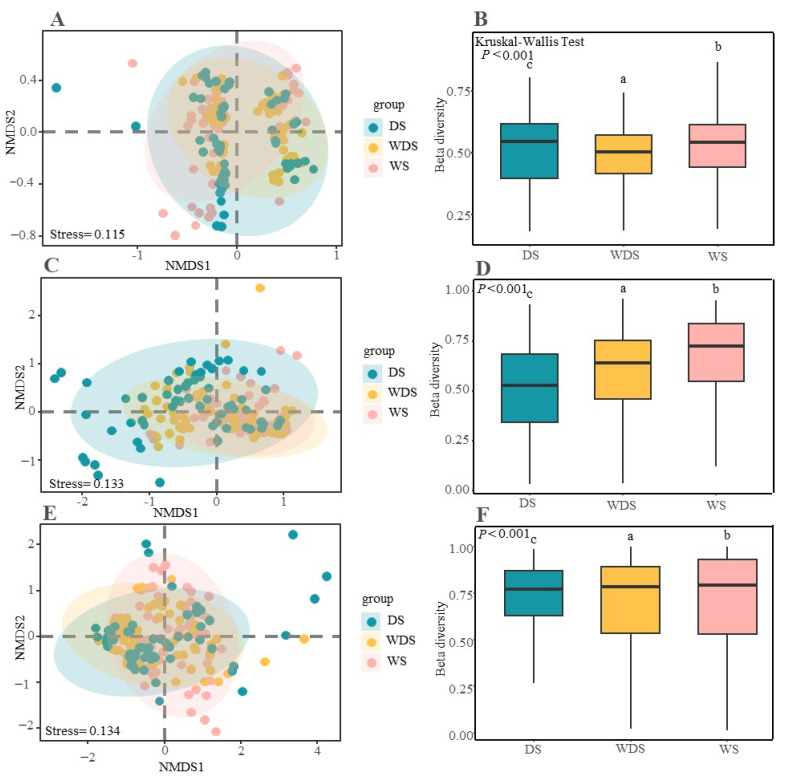
Beta diversity of the microbial community in the different flooding states. NMDS analysis showing the microbial communities of soil bacteria (**A**), fungi (**C**), and archaea (**E**), which were estimated on the basis of a Bray–Curtis distance matrix of all 180 soil samples. Data with different letters indicate a significant difference between the flooding treatments (*p* < 0.05, multiple comparisons were based on Kruskal–Wallis test). Differences in beta diversity among the bacteria (**B**), fungi (**D**), and archaea (**F**), were estimated based on a Bray–Curtis distance matrix of soil samples. Lowercase letters indicate significant differences among flooding conditions (*p* < 0.05).

**Figure 4 microorganisms-13-00154-f004:**
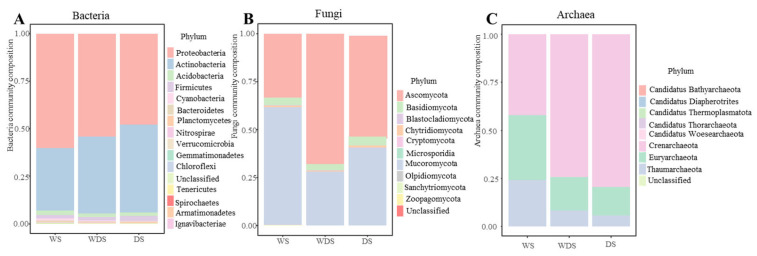
Composition of the bacterial (**A**), fungal (**B**), and archaeal (**C**) communities at the phylum level in 180 samples. Treatment names are given below the graph. The percentages of the classified sequences are shown.

**Figure 5 microorganisms-13-00154-f005:**
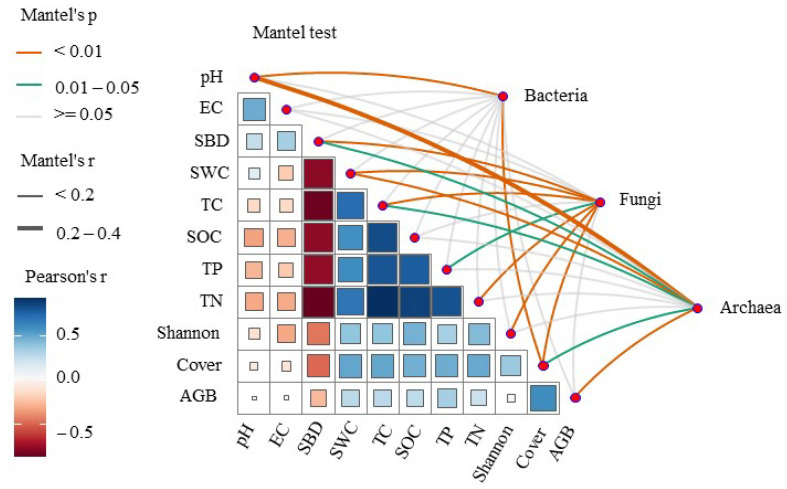
Mantel test was conducted to assess the relationship between microbial communities and environmental factors. pH, soil pH; EC, electrical conductivity; Cover, plant cover; SBD, soil bulk density; SWC, soil water content; TC, total nitrogen; SOC, soil organic carbon; TP, total phosphorus; TN, total nitrogen; Shannon, plant Shannon diversity; AGB, aboveground biomass.

**Figure 6 microorganisms-13-00154-f006:**
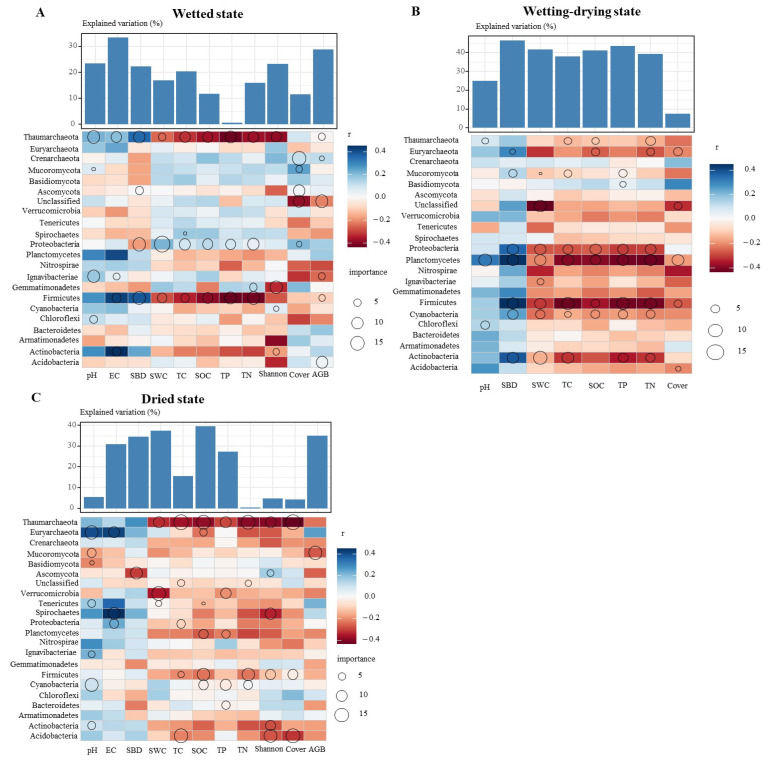
The contribution of the core microbes to soil nutrients was measured via the correlation of the main categories of microbes (bacteria community) and the best multiple regression model (in different flooding conditions, (**A**) is Wetted state; (**B**) is Wetting-drying state; (**C**) is dried state). Different circle sizes represent the variable’s importance. (**A**) The colors represent Spearman correlations. pH, soil pH; EC, electrical conductivity; SBD, soil bulk density; SWC, soil water content; TC, total nitrogen; SOC, soil organic carbon; TP, total phosphorus; TN, total nitrogen; Shannon, plant Shannon diversity; Cover, plant cover; AGB, aboveground biomass.

### 3.3. The Relationships of Soil Microbes with the Plant Community and Soil Environmental Factors

Piecewise SEM was performed to identify the different pathways (direct and indirect) by regulatory factors that impacted the various states of bacterial, fungal, and archaeal diversity. The Shannon index and SWC had negative effects on bacterial diversity. Different flooding states indirectly affected bacterial diversity via SWC. The flooding state and Shannon indices were positively associated with fungal diversity, and the TP and pH indices were negatively correlated with fungal diversity. The Shannon index and state had significantly negative effects on archaeal diversity, and the flooding state directly promoted the SBD and AGB (Figure 7).

## 4. Discussions

### 4.1. The Effects of Different Flooding Conditions on the Microbial Community

Wetlands are specific ecosystems that are frequently or continually flooded with water, including marshes, bogs, swamps, and fens [34]. Wetlands are among the largest reservoirs of terrestrial carbon, which include organic matter and decayed herbaceous materials [35]. Soil moisture is expected to be the main consequence of the soil microbial community in various ecosystems (e.g., grassland or forest systems) [36,37]. In addition, arid areas will increase in many global regions, and most semiarid and arid regions will generally face desertification in the future [38]. Soil microbes widely exist in a variety of ecosystems, which range from deserts, with limited rainfall and spread, to forests and wetlands, making it difficult to generalize their response to soil moisture [39]. Drought has long-term effects on the soil microbial community, which further shifts local vegetation to more drought-resistant plant species, resulting in the selection of various root-associated microbes [40]. Drought significantly reduces microbial diversity and abundance by reducing vegetation diversity and the organic carbon content of soil. Under extreme conditions, a variety of bacterial lineages exist in arid soils, such as *Actinobacteria*, *Gemmatimonadetes*, *Proteobacteria*, *Chloroflexi*, and *Firmicutes* [41,42]. However, we found that the Shannon diversity of bacteria in the dry state was greater than that in the wetting state. *Xerotolerant* bacteria survive in harsh ecological niches, such as the Atacama and Antarctic deserts, fissures, pores, and cracks on rocky surfaces, and these bacteria are highly tolerant to drought [43]. Our results revealed that the number of fungi in the DS treatment was significantly greater than that in the WDS and WS treatments. Compared with bacteria, fungal communities are less affected by drought stress (e.g., Arbuscular mycorrhizal fungi), and fungi are more resistant to drought because they have the ability to form hyphal networks and spores. Moreover, these hyphal networks provide nutrients and water for plants, thereby improving the drought tolerance of vegetation [44]. In recent years, additional studies have confirmed that AMF are also widely present in a variety of wetland systems around the world [45,46]. However, with a variety of adverse factors (e.g., abiotic factors and biotic factors), the colonization of AMF in wetland plant roots is still low [45,47]. The colonization of AMF is affected by water regimes, which are relatively high in wetlands, and when the oxygen content is relatively low, where AMF cannot easily survive and colonize plant roots [48]. Second, the soil of wetlands lacks the available oxygen that fungi need to survive. A multiyear field experiment revealed that soil fungi are less sensitive to drought than soil bacteria are in grasslands, which are more drought-tolerant and maintain the cycling of carbon and nitrogen under water-scarce conditions [49]. In our study, the Shannon diversity of archaea had the opposite trend compared with that of bacteria, and that of WS was significantly greater than that of DS and WDS. Archaea constitute the third domain after bacteria and eukaryotes, and constitute an unnecessary part of the soil microbial biomass and community, which results in extreme or anaerobic environments [42]. For example, euryarchaeotal methanogenesis is typically considered the primary process in anaerobic environments [50]. A majority of archaea contribute to the biogeochemical cycles of carbon, nitrogen, and hydrogen [51]. The environment for archaea assembly was not consistent, and there were different ecological niches between maize field with low water content and rice field with high water content. However, the abundance or community of archaea was not different with altitude in alpine forests [52,53]. Recent observational studies have revealed that greater microbial diversity increases multiple ecosystem functions at the global scale and that biodiversity–ecosystem functional relationships are connected mainly with α- and β-diversity, community construction, and biodiversity assemblages [42]. Our results revealed that different flooding levels changed the beta biodiversity of bacteria, fungi, and archaea, and shaped their structure, highlighting the vital importance of soil water regimes. A previous study revealed that the disappearance of ubiquitous species from certain locations led to lower α-diversity, but dissimilar environmental conditions resulted in the selection of different species, resulting in greater β-diversity (subtractive heterogenization) [54]. Proteobacteria was the dominant species in our results; it is present in a variety of complex ecological environments (nutritional deficiency), especially in wetland soil (consistent with our results), and is intimately connected with the nitrogen cycle and N_2_O emissions during global change [55]. Actinomyces are aerobic heterotrophic microbes that exist in oxygen-rich environments [56], and the abundance of *Actinomyces* was greater in WDS and DS than in WS (with relatively high soil moisture) in our study.

Archaea are related to the element circles of carbon, nitrogen, and sulfur [57]. The role of archaea in the ecological cycle should not be underestimated, and some archaea can perform nitrogen absorption and nitrogen fixation. In the carbon cycle, methane-producing archaea act as decomposers [58]. Archaea live mostly in extreme environments and are widely distributed in various ecosystems [59]. They are also found in high-temperature and freezing environments. *Crenarchaeota* can grow at high temperatures, some of which are extremely thermophilic microbes, and the optimal growth temperature is more than 85 °C [60]. In our study, we found that *Crenarchaeota* were concentrated mainly in WDS and DS, which have relatively little water, rather than in flooded areas (WS). Wetlands are among the largest natural sources of methane on earth, and *Euryarchaeota* are methanogens that exist widely in wetlands [61]. In our study, *Euryarchaeota* were more abundant in WS than in WDS and DS. *Thaumarchaeota* are more common in environments with high moisture conditions, such as the ocean, and at our sample sites, *Thaumarchaeota* were also more concentrated in WS than in WDS and WD.

### 4.2. Soil Physicochemical Properties Drive the Microbial Community

The NMDS results revealed that the soil’s bacterial, fungal, and archaeal community structures significantly differed among the different flooding conditions (Figure 3). Previous studies have indicated that the soil water content and soil pH affect the microbial community, which strongly influences the distribution pattern of the soil microbial community [62,63]. Soil pH is a significant factor that drives the community structure of soil bacteria in the surface soil layer (0–20 cm). The Mantel analysis results revealed that soil pH was the crucial driver of the soil bacterial community structure in the Huihe wetlands in our study (Figure 5). Many previous studies have also demonstrated that soil pH is the driving factor controlling the soil bacterial community structure [64]. A recent meta-analysis also indicated that soil pH is an important predictive factor for microbial diversity (alpha) among many different abiotic factors and that the response ratio (RR) for soil microbial richness and the Shannon index increase with increasing soil pH [65]. In our study, we found that soil pH affected the bacterial community structure, and random forest analysis and heatmaps further demonstrated the contributions of the soil physicochemical property differences to microbial species. The soil pH, SBD, plant Shannon index, and plant AGB highly contributed to the bacterial community (Table 1), which was similar to the results for wetlands in the Changdu area, Tibet [66]. In alpine wetlands, surface soil pH has a negative influence on soil bacterial diversity but a positive influence on fungal α-diversity [67]. The possible reason was the obvious difference in the tolerance of bacteria and fungi to soil pH: bacteria generally prefer neutral conditions and are sensitive to soil pH, whereas fungi prefer relatively acidic conditions [68]. Moreover, soil pH enhanced the network interaction between bacterial and archaeal communities in the alpine grassland of the Qinghai–Tibet Plateau, which was similar to our results that pH affects the archaeal community structure [69]. Second, soil pH is the most important environmental variable for predicting the network-level topological characteristics of soil microbial symbiotic networks [70]. The correlation between soil microbiota species increased with increasing pH (5.17~8.92), and the network was most stable at neutral pH [71]. Therefore, soil pH is a key environmental filter affecting the potential associations and ecological characteristics of the soil microbiota in the Tibetan alpine steppes [72]. Soil fungi, especially soil mycorrhizal fungi, are closely related to the soil water content, which can play an important role in the water acquisition and drought tolerance of plants [73]. Research on alpine riparian wetlands has shown that soil moisture is a prominent predictor of the AMF community in the far bank [74]; in our study, soil moisture was also found to affect the soil fungal community structure. The SWC, SBD, TN, and TC contributed strongly to the fungal community, and the soil moisture varied in the riparian wetland. Soil moisture may become a limiting factor for wetland ecosystems [75]. Water restriction directly affects the type of AMF, and different soil moisture levels affect the colonization of AMF [74], the distribution and growth of mycelia, and even the formation of different and specific combinations of AMF [76]. Moreover, the oxygen demand of fungi is high, and soil aeration affects the diversity and community structure of fungi; therefore, soil bulk density affects soil permeability and further affects soil fungi [77]. Wetlands store about one-third of the soil’s carbon, which plays a crucial role in the global carbon cycle [78]. Soil fungi (especially AMF) regulate the distribution and stabilization of photosynthetic carbon from plants to soil, further affecting the global carbon cycle [78]. Nitrogen deposition is considered a serious threat to terrestrial ecosystems because it has many negative effects on ecosystem structure and function, such as reducing plant diversity and soil microbes [79]. The decomposition of soil organic matter by fungi depends on the availability of soil nitrogen. Specifically, nitrogen deposition increases the availability of soil nitrogen, leading to soil acidification and affecting fungal communities [79]. Simulated N deposition increases the fungal richness and biomass in low N depositional plots and decreases the fungal richness in high N depositional plots [80]. A decrease in fungal community richness or biomass leads to soil C cycling, and a decrease in mycelia reduces the formation of large aggregates and thus reduces soil C sequestration [80].

### 4.3. Mechanisms of the Effects of Soil and Plant Properties on Soil Microbes

The physical and chemical properties of vegetation and soil do not affect microorganisms independently but rather indirectly or directly together, so their relationships are more complicated [81]. We used structural equation models to further demonstrate the effects of the soil physicochemical properties of vegetation and soil on bacteria, fungi, and archaea. WS, WDS, and DS had no influence on bacterial diversity. WS, WDS, and DS affected the soil’s fungal and archaeal diversity. A study of the wetlands in the Yellow River delta revealed that the microbe abundances of *Phragmites australis* and *Tamarix chinensis* were relatively high [82]. Variation in vegetation further affects soil microbial diversity and function [83,84]. Plants depend on microorganisms to absorb and utilize elements such as nitrogen (N) and phosphorus (P) in the soil, and microbes benefit their hosts by increasing the absorption of nutrients and water and resisting other pathogens and environmental stress [85,86]. However, the physicochemical properties, organic matter content, and the nutrient composition of soil vary, which leads to differences in the soil microorganisms in wetlands [87,88]. Therefore, plant diversity can be used to predict the diversity of soil microbial communities. Soil properties affect the soil microbial community and composition, and the relationships between soil properties and soil microorganisms are mutual [89,90]. Microbe–soil interactions require causality and feedback networks in which previously selected microbes drive environmental change [91], so microbially driven changes in the soil properties subsequently shape microbial community composition and activity [92,93].

## 5. Conclusions

This research revealed the potential relationships among plants, soil physicochemical properties, and the soil microbial community composition in the Huihe wetland. The research results provide significant guidance for the ecological restoration of the Huihe wetland. The results revealed that fungi and archaea were significantly different in WS, DS, and WDS conditions, while the flooding condition did not affect the bacterial biodiversity. Community compositions of bacteria, fungi, and archaea were different under WS, DS, and WDS. The soil pH affected the bacterial and archaeal communities, and the SBD, SWC, TC, and TN affected the fungal community. These results will help to better understand the internal mechanism among plants, soil microbial communities, and soil physicochemical properties in the Hui River wetland, guide the restoration of wetland plants and the soil environment, and maintain the stability of the Huihe Nature Reserve.

## Figures and Tables

**Figure 1 microorganisms-13-00154-f001:**
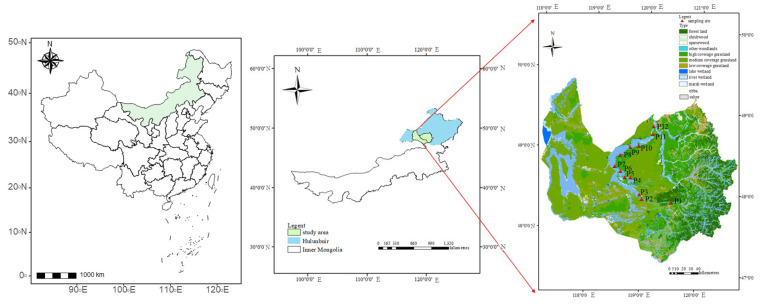
The location of the sampling area.

**Figure 2 microorganisms-13-00154-f002:**
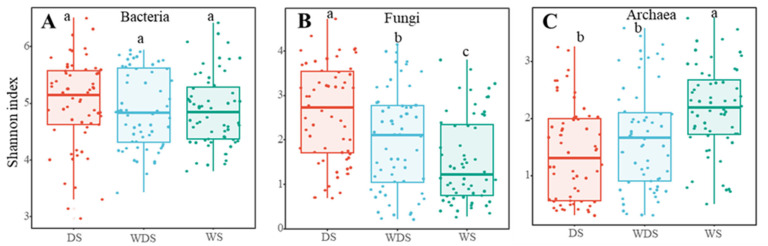
Shannon index (alpha diversity) of bacterial (**A**), fungal (**B**), and archaeal (**C**) communities in different flooding conditions (DS, WDS, and WS). Lowercase letters indicate significant differences among flooding conditions (*p* < 0.05). Red means Shannon index (alpha diversity) of bacterial under DS (Drying state), blue means Shannon index (alpha diversity) of bacterial under WDS (Wetting-drying state), green means Shannon index (alpha diversity) of bacterial under WS (Wetting state).

**Figure 7 microorganisms-13-00154-f007:**
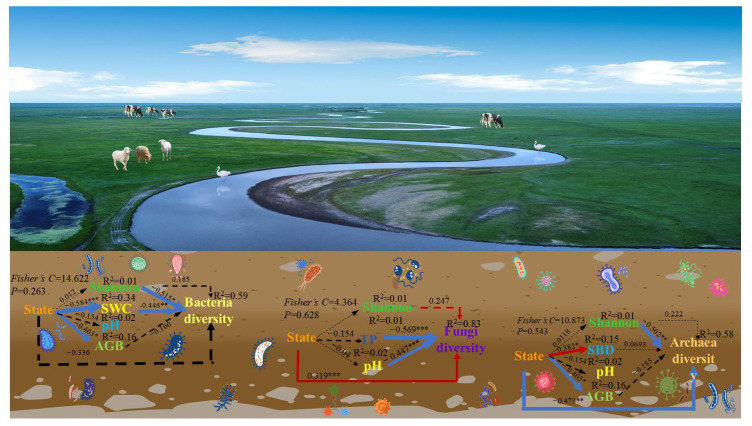
Piecewise SEM was further utilized to analyze the pathways through which the flooding state impacted soil bacterial, fungal, and archaeal diversity via plant and soil properties at the Huihe Nature Reserve. Composite variables were generated to represent the soil properties, plants, and microbes. Significant effects were defined as follows: * 0.01 < *p* < 0.05; ** 0.001 < *p* < 0.01; *** *p* < 0.001. Our results showed that the responses of soil bacterial diversity, fungal diversity, and archaeal diversity were influenced by multiple direct and indirect effects on plant and soil properties.

**Table 1 microorganisms-13-00154-t001:** Effects of wetland of biotic and abiotic properties. Means within a row with different lowercase letters (a, b, c) differ from each other (*p* < 0.05). Bold values represent significant differences among treatments. WS: wetted state, WDS: wetting-drying state, DS: dried state. TC: soil total carbon; TN: soil total nitrogen; TP: soil total phosphorus; SOC: soil organic carbon; SBD: soil bulk density; SWC: soil water content; AGB: aboveground biomass.

Variables	WS	WDS	DS
TC (g·kg−1)	**48.74 ± 6.126 a**	**46.725 ± 6.099 a**	**30.383 ± 2.558 b**
TN (g·kg−1)	3.395 ± 0.366 a	3.214 ± 0.381 a	2.464 ± 0.193 a
TP (g·kg−1)	0.5 ± 0.046 a	0.572 ± 0.066 a	0.459 ± 0.039 a
SOC(g·kg−1)	30.788 ± 3.237 a	32.078 ± 4.37 a	23.421 ± 2.251 a
pH	8.485 ± 0.17 a	8.523 ± 0.197 a	8.319 ± 0.228 a
SWC (%)	**48.644 ± 2.313 a**	**40.109 ± 2.603 b**	**31.965 ± 1.969 c**
SBD (g·cm−3)	**1.181 ± 0.056 a**	**1.252 ± 0.053 a**	**1.334 ± 0.027 b**
Aboveground biomass (g·m−2)	**168 ± 15.29 b**	**170 ± 30.48 b**	**135 ± 7.97 b**
Shannon diversity	**230 ± 16.15 b**	**221 ± 31.75 b**	**286 ± 10.41 a**

## Data Availability

The original contributions presented in this study are included in the article. Further inquiries can be directed to the corresponding authors.

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
