# Peer review of "Different Flooding Conditions Affected Microbial Diversity in Riparian Zone of Huihe Wetland"

_microorganisms, 2025, doi:10.3390/microorganisms13010154_

Round 1
Reviewer 1 Report
Comments and Suggestions for Authors
Dear Authors,
Your manuscript is interesting because it shows the different flood conditions affecting the microbial diversity in the coastal zone of the Huihe wetland. The layout of the article is standard for research papers. However, Authors should read the instructions on how to properly prepare a manuscript for the "Microorganisms" journal.
General comments:
The title and abstract correspond to the content of the manuscript. However, I suggest that you make minor changes to the keywords as instructed below (L 2-3 and L 29). The introduction provides the most important information about wetland ecosystem, specific soil microbial communities and interactions between vegetation community structure and soil microbes. The hypotheses and objectives of the study were well explained. The research methods are incomplete and this chapter needs to be completed. The results were well characterized. The discussion was well described. However, the summary should be improved and the conclusions should be written clearly. References are adequate to the content of the manuscript.
Detailed comments
L 2-3 and L 29: Some keywords are repeated from the manuscript title. Please provide others.
L 33: “(Cai et al., 2021)” In the text, reference numbers should be placed in square brackets [ ], and placed before the punctuation; for example [1], [1–3] or [1,3]. Prepare a document according to the guidelines for Microorganisms https://www.mdpi.com/journal/microorganisms/instructions#references
- L 57: “SOC” Acronyms/Abbreviations/Initialisms should be defined the first time they appear in each of three sections: the abstract; the main text; the first figure or table (https://www.mdpi.com/journal/microorganisms/instructions)
L 112-113: “(wetted state, wetting–drying state and dried state)” - Here you can add abbreviations, it will be a good introduction to your research.
L 121-122: WBR should be cited and included in the references.
L 147-148: Save the formula using the dedicated equation editor (according to the instructions for authors https://www.mdpi.com/journal/microorganisms/instructions).
L 225-229 and others: Microbial community names should be in italics.
L 249: “SWC” What does this abbreviation mean? Why is this parameter not described in Chapter 2?
L 250: “AGB” or L 328 : “AMF”….. What do these abbreviation mean? Please check the entire manuscript carefully and explain all abbreviations. Also make sure that the methodology of the parameters discussed is described.
L 274: fix font size.
L 461: The conclusions are too general and do not respond to the stated hypotheses and research objectives.
L 487: References must be numbered in order of appearance in the text (https://www.mdpi.com/journal/microorganisms/instructions#references)
I do not feel qualified to evaluate the English language and style.
Good luck!
Sincerely yours
Reviewer
Author Response
- L 2-3 and L 29: Some keywords are repeated from the manuscript title. Please provide others.
Response: Thank you for your valuable comments. We had provided other keywords in L29.
- L 33: “(Cai et al., 2021)” In the text, reference numbers should be placed in square brackets [ ], and placed before the punctuation; for example [1], [1–3] or [1,3]. Prepare a document according to the guidelines for Microorganisms https://www.mdpi.com/journal/microorganisms/instructions#references
Response: Thank you for pointing out the error, we had corrected all the reference form in the revised version.
- L 57: “SOC” Acronyms/Abbreviations/Initialismsshould be defined the first time they appear in each of three sections: the abstract; the main text; the first figure or table (https://www.mdpi.com/journal/microorganisms/instructions)
Response: Thank you for pointing out the error, which has been corrected in the revised version (Line 55).
- L 112-113: “(wetted state, wetting–drying state and dried state)” - Here you can add abbreviations, it will be a good introduction to your research.
Response: Thank you for pointing out the error, we used abbreviations in the revised version (Line 109).
- L 121-122: WBR should be cited and included in the references.
Response: Thank you for pointing out the error, WBR had provided reference in the revised version (Line 119).
- L 147-148: Save the formula using the dedicated equation editor (according to the instructions for authors https://www.mdpi.com/journal/microorganisms/instructions).
Response: Thank you for pointing out the error, formula used equation editor in the revised version (Line 119).
- L 225-229 and others: Microbialcommunity names should be in italics.
Response: Thank you for pointing out the error, all the name of microbes had been corrected in the revised version (Line 119).
- L 249: “SWC” What does this abbreviation mean? Why is this parameter not described in Chapter 2?
Response: Thank you for pointing out the error, SWC is soil water content and it had been corrected in the revised version (Line 153).
- L 250: “AGB” or L 328 : “AMF”….. What do these abbreviation mean? Please check the entire manuscript carefully and explain all abbreviations. Also make sure that the methodology of the parameters discussed is described.
Response: Thank you for pointing out the error, we had checked the entire manuscript carefully and explain all abbreviations in the revised version (Line 153).
- L 274: fix font size.
Response: Thank you for pointing out the error, we had fix font size in the revised version (Line 153).
- L 461: The conclusions are too general and do not respond to the stated hypotheses and research objectives.
Response: Thank you for pointing out the error, we had improved conclusions in the revised version (Line 153).
- L 487: References must be numbered in order of appearance in the text (https://www.mdpi.com/journal/microorganisms/instructions#references)
Response: Thank you for pointing out the error, all references had been corrected in the revised version.

Reviewer 2 Report
Comments and Suggestions for Authors
This manuscript is valuable for science, especially ecology, and should be published in Microorganisms. It provides many very detailed data for understanding the effects of different hydrological conditions on microbial communities in Huihe Nature Reserve (China). The methodological aspects have been well thought out. 12 plots along the Huihe River were selected and on each of them three variants were chosen, depending on the moisture: a wetted state (WS), a wetting-drying state (WDS) and a dried state (DS). Every condition included 5 random replications. The total number of plant species, coverage of plants and aboveground biomass were determined in 180 squares. Surface soil samples were collected for physical, chemical and metagenome analyses of soil. The only unclear element is this: In Material and Methods (chapter 2.5.2.) it is stated that "To identify the species, present in the samples, Kraken2 was used ...". In „Results”, the results regarding microorganisms for the phyla are presented. This requires explanation. The research provided numerous detailed results on the diversity of bacteria, fungi and archaea depending on moisture conditions, plant communities, physical and chemical properties of the soil. The results are presented clearly and are supported by statistical analyses. The discussion is very interesting. The conclusion is correct. The manuscript is carefully prepared. Only minor corrections should be made (see Remarks).
Remarks
Line 16 "did not significantly affect bacterial diversity", Line 18 "Significant differences were found in the beta-diversity of bacterial ... community" - it would be good to clarify this text
Line 33 (Cai et al., 2021) should rather be given the number
Line 155 the methods of Ye - it is unclear, is this Je et al. cited in References under numbers 77 and 78?
Line 206 - factors(Yang - add a space
Line 233 it should be (DS., WDS, WS - different flooding states).
Line 235 Figure 3 explanation for B, D, F should be completed
Line 227-228 this text is not precise, Ascomycota, Basidiomycota… - this is a phylum (as given in line 241)
Line 278 (at the OTU level) there is no information about it in the text, only in Figure 5
Line 328 the abbreviation AMF (probably Arbuscular Micorrizal Fungi) should be explained
Line 393 with increasing soil pH - is there an upper limit to which this happens?
Line 443 "WS, WDS, and DS had no influence on bacterial diversity". This should be compared with the text in Line 18 and the text should be clarified
Line 445-446 Phragmites australis .. Tamarix chinensis – it should be in italic
Line 469 ‘but significantly affected fungal and archaeal diversity’ – this is a repetition of the text from line 465
Line 552 no bibliographic data
Line 554 CATENA - no bibliographic data
Line 583 text requires explanation
Line 637 Rhododendron nitidulum - it should be italic
Author Response
- Line 16 "did not significantly affect bacterial diversity", Line 18 "Significant differences were found in the beta-diversity of bacterial ... community" - it would be good to clarify this text.
Response: Thank you for pointing out the error, which has been corrected in the revised version (Line 16).
- Line 33 (Cai et al., 2021) should rather be given the number.
Response: Thank you for pointing out the error, we repeated all the reference according the rules in the revised version.
- Line 155 the methods of Ye - it is unclear, is this Je et al. cited in References under numbers 77 and 78?
Response: Thank you for pointing out the error, which has been corrected in the revised version.
- Line 206 – factors (Yang - add a space)
Response: Thank you for pointing out the error, which has been corrected in the revised version (Line203).
- Line 233 it should be (DS., WDS, WS - different flooding states).
Response: Thank you for pointing out the error, which has been corrected in the revised version (Line230).
- Line 235 Figure 3 explanation for B, D, F should be completed.
Response: Thank you for pointing out the error, which has been corrected in the revised version (Line237).
- Line 227-228 this text is not precise, Ascomycota, Basidiomycota… - this is a phylum (as given in line 241)
Response: Thank you for pointing out the error, which has been corrected in the revised version (Line225).
- Line 278 (at the OTU level) there is no information about it in the text, only in Figure 5
Response: Thank you for pointing out the error, which has been corrected in the revised version (Line280).
- Line 328 the abbreviation AMF (probably Arbuscular Micorrizal Fungi) should be explained.
Response: Thank you for pointing out the error, which has been corrected in the revised version (Line327).
- Line 393 with increasing soil pH - is there an upper limit to which this happens
Response: There are different pH in different places, which ranges 4.5-8.5 generally.
- Line 443 "WS, WDS, and DS had no influence on bacterial diversity". This should be compared with the text in Line 18 and the text should be clarified
Response: Thank you for pointing out the error, which has been corrected in the revised version (Line 18).
- Line 445-446 Phragmites australis. Tamarix chinensis – it should be in italic
Response: Thank you for pointing out the error, which has been corrected in the revised version (Line437).
- Line 469 ‘but significantly affected fungal and archaeal diversity’ – this is a repetition of the text from line 465
Response: Thank you for pointing out the error, which has been corrected in the revised version (Line459)
- Line 552 no bibliographic data
Response: Thank you for pointing out the error, which has been corrected in the revised version (Line 603)
- Line 554 CATENA - no bibliographic data
Response: Thank you for pointing out the error, which has been corrected in the revised version (Line 29)
- Line 583 text requires explanation
Response: Thank you for pointing out the error, which has been corrected in the revised version (Line 578)
- Line 637 Rhododendron nitidulum - it should be italic
Response: Thank you for pointing out the error, which has been corrected in the revised version (Line 29)
